# Least Square Calibration for Peer Reviews

**Sijun Tan***
Department of Computer Science
University of Virginia
Charlottesville, VA 22903
st8eu@virginia.edu

**Jibang Wu***
Department of Computer Science
University of Virginia
Charlottesville, VA 22903
jw7jb@virginia.edu

**Xiaohui Bei**
School of Physical and Mathematical Sciences
Nanyang Technological University
Singapore 637371
xhbei@ntu.edu.sg

**Haifeng Xu**
Department of Computer Science
University of Virginia
Charlottesville, VA 22903
hx4ad@virginia.edu

## Abstract

Peer review systems such as conference paper review often suffer from the issue of miscalibration. Previous works on peer review calibration usually only use the ordinal information or assume simplistic reviewer scoring functions such as linear functions. In practice, applications like academic conferences often rely on manual methods, such as open discussions, to mitigate miscalibration. It remains an important question to develop algorithms that can handle different types of miscalibrations based on available prior knowledge. In this paper, we propose a flexible framework, namely *least square calibration* (LSC), for selecting top candidates from peer ratings. Our framework provably performs perfect calibration from noiseless linear scoring functions under mild assumptions, yet also provides competitive calibration results when the scoring function is from broader classes beyond linear functions and with arbitrary noise. On our synthetic dataset, we empirically demonstrate that our algorithm consistently outperforms the baseline which select top papers based on the highest average ratings.

## 1 Introduction

Peer review has been essential to the advancement of academic research. In a typical conference peer review process, the submitted papers are assigned to a set of external reviewers and will receive cardinal scores from them. The final decision of which papers get accepted to the conference is usually determined by the average score received for each paper. However, during such a process, different reviewers may have different standards and biases in their evaluations. For example, the same paper may receive a high score (e.g., 8 out of 10) from a lenient reviewer and a low score (e.g., 2 out of 10) from some strict reviewer. As a result, the average score may not always reflect the true quality of a paper. This issue is known as *miscalibration*, and is a prevalent problem in many other online review systems as well such as TripAdvisor or Yelp. In order to ensure a fair and unbiased outcome, it is crucial to the concept and to the success of a peer reviewing system to have an efficient and effective calibration algorithm.

There has been a body of works attempting to address the miscalibration issue in peer reviews. Most of these works can be categorized into two directions. The first line of works relies on making simplifying assumptions on the form of miscalibration. For example, [19, 5, 10, 16, 27]

---
*Equal Contribution

35th Conference on Neural Information Processing Systems (NeurIPS 2021).

model the miscalibration as one-parameter additive bias for each reviewer; [19, 21] assume each reviewer's scoring function to be a linear function. Many of these works are able to accurately learn the unknown parameters of the miscalibration due to its simple structure. However, these strong structural assumptions on the miscalibration are often too strong for these results to be widely applicable. The second line of research [20, 9, 13, 17, 18, 3] allows the miscalibration to be arbitrary, and only relies on ordinal ranking information obtained from cardinal scores to calibrate. A recent work [26] bridges these two lines of research by showing that even when the reviewers' functions are arbitrary, there exist cardinal score estimators that perform better than any ordinal ranking estimators. However, their work is still pertained to worst case assumptions, and is restricted to the scenario where each reviewer only reviews a pair of items. It remains an important open problem to design calibration models and algorithms that can work with weaker simplifying assumptions (e.g non-linear functions) in a practical scenario.

**Summary of Contributions.** In this work, we propose a new optimization-driven unsupervised learning framework, *least square calibration* (LSC). As the name suggests, the framework leverages the natural and powerful least square method for handling different types of miscalibrations in peer reviews. A key strength of our framework is its *flexibility* in incorporating various levels of prior knowledge, which consequently results in different qualities of calibration. Specifically, we capture the system's prior knowledge about reviewers' scoring as a *hypothesis class* of scoring functions. Depending on how much the system knows about reviewers, the hypothesis class can be as "rough" as all monotone functions, or as specific as linear functions, convex/concave functions, Lipschitz continuous functions, or even their mixtures. Our framework can be tailored to accommodate all these hypothesis classes and moreover, cope with the reviewers' noisy perception of true item qualities.

We demonstrate the effectiveness of the LSC framework both theoretically and empirically. We start our theoretical analysis from the hypothesis class of *linear* scoring functions. When reviewers' scores are generated by *arbitrary* linear functions of true item qualities *without* noise, our main result is a necessary and sufficient characterization about when LSC can (essentially) perfectly recover true item qualities. Interestingly, it turns out that perfect recovery is fully captured by a novel graph connectivity notion for a carefully constructed reviewer graph, which we coin *double connectivity*. This result reveals useful insights about what kind of reviewer-item assignments may be good for calibration, which is further justified in our experiments. When reviewers perceive item qualities with noises, we show how to reduce the LSC framework to a tractable convex optimization program when scoring functions are linear. We then extend this technique to accommodate other function classes including monotone, convex, or Lipschitz continuous functions. Our theoretical analysis is concluded with a discussion about the connection of LSC to other well-known problems such as regression and matrix seriation.

Empirically, we test the performance of LSC via extensive experiments. We show that LSC consistently outperforms previous methods in various problem setups. The experiments also demonstrate that the calibration quality of LSC improves as our prior knowledge improves and LSC is robust to reasonable amount of misspecified prior knowledge. We hope these encouraging findings can motivate more future studies about how to identify more accurate class of reviewers' scoring functions.

## 2  Problem Formulation

In this section, we will first formally describe the peer review problem and then introduce our calibration approach. We use conference peer review as an illustrative example, though our model and methods can be trivially generalized to many other peer review systems.

**The Peer Review Problem.** In a peer review process such as an academic conference, there are $N$ items and $M$ reviewers. By convention, we use $[N] = \{1, \cdots, N\}$ to denote the set of items and $[M] = \{1, \cdots, M\}$ to denote the set of reviewers. We assumes that each item has an inherent ground truth quality $x^*(i)$. Throughout the paper, the superscript notation "$*$" will always be used to denote *true* qualities. For simplicity, we assume that there are no ties and the item qualities are all distinct from each other. Each reviewer $j$ reviews a subset of the items $I_j \subseteq [N]$. For now, we allow $I_j$ to be an arbitrary subset, which can even be a singleton or the entire set of items $[N]$. We will specify later under what conditions about $\{I_j\}_{j\in[M]}$ our algorithms will work. We denote $I_j^\ell \in [N]$ as the *index* of the $\ell$'th item reviewed by reviewer $j$.

**Scoring Functions and Score Generations.** Each reviewer $j \in [M]$ is characterized by a monotone *scoring function* $f_j : \mathbb{R} \to \mathbb{R}$. Like most previous works in this space [10, 21, 5, 16, 27], we assume that reviewers' scores are generated by applying her scoring function to a *noisy* perception of the item's true quality. Formally, reviewer $j$'s score about item $I_j^\ell$, denoted by $y_j^\ell$, is generated as follows

$$\text{Score Generation:} \quad y_j^\ell := f_j(x^*(I_j^\ell) + \epsilon_j^\ell) \tag{1}$$

where $f_j$ is reviewer $j$'s scoring function and $\epsilon_j^\ell$ is an independent zero-mean noise. For convenience, we always assume the scores for each reviewer $j$ is ordered increasingly such that $y_j^1 < y_j^2 \cdots < y_j^{|I_j|}$.

**Design Goal.** We are interested in designing an item selection algorithm. The algorithm takes as input the item assignment $\{I_j\}_{j\in[N]}$ for a set of items $[M]$ and a set of reviewers $[N]$, the reviewers' scores for each of their assigned item $\{y_j^\ell\}_{j\in[M],\ell\in[I_j]}$, and an additional threshold parameter $n \leq N$. The goal of the algorithm is to output a set $S$ of $n$ items that best matches the top $n$ items according to the (unknown) true qualities $x^*$. In the experimental section, we will describe multiple metrics to evaluate the performance of our algorithm.

**Restricted Classes of Scoring Functions.** Note that when defining reviewers' scoring functions, we do not make any assumptions on the properties of these functions other than monotonicity. This is to ensure that our framework is flexible and can handle a wide range of problems.

However, in many situations, the designer may have some additional prior knowledge and infer more structures about the scoring functions. The reason of assuming some restricted class of scoring functions is multifold. First, such functions may be a good proxy or approximation of the true scoring functions based on our prior knowledge. Second, it may help us to avoid the issue of *overfitting*[2] since allowing arbitrary scoring functions may be able to fit the review scores well but leads to poor generalization and make the result prone to perception noise. For example, many previous works [21, 10, 5, 16] have focused on the basic model of *linear* scoring functions, defined as follows.

**Definition 2.1** (Linear Scoring Function). *A linear scoring function of reviewer $j$ has two parameters: $a_j \geq 0$ and $b_j \in \mathbb{R}$ and is of the form $f_j(\widetilde{x}(= x + \epsilon)) = a_j\widetilde{x} + b_j$.*

In addition to linear scoring functions, one may also consider broader class of scoring functions when less prior knowledge is available or when we want to make less stringent assumptions. For example, our framework can also handle scoring functions that are convex or concave or a mixture of these functions.[3] However, we emphasize that it is *not* our purpose to justify which of these classes of scoring functions are more realistic since this answer would depend on the concrete domains of the problem. Instead, our goal is to develop a generic learning framework that can integrate any of such prior knowledge to improve the calibration. That being said, we believe it is an important future research direction to design heuristics, regularizations and other methods to choose the hypothesis class for better calibration results as well as to understand the potential issue of overfitting for hypothesis class that are too broad.

## 3 The Optimization-Driven Framework

Central to our approach is a framework to recover the underlying item quality $\mathbf{x}$ that can properly "explain" the observed review scores given the assumptions about the scoring function structures. Note that our recovered qualities may be up to some constant shifting or re-scaling from the true qualities. However, as long as the shifting and re-scaling preserve monotonicity, this will not affect the accuracy of our selection of the best $n$ items.

Following the notation in machine learning, let $\mathcal{H}$ denote a *hypothesis class* of scoring functions. For example, for linear scoring functions, the hypothesis class is $\mathcal{H} = \{f : f(x) = ax + b \text{ for some } a \geq 0, b \in \mathbb{R}\}$. We denote this linear scoring function class as $\mathcal{H}_L$ for convenience. More generally, $\mathcal{H}$ may be any class of functions depending on designer's prior knowledge of these scoring functions.

Given a hypothesis class $\mathcal{H}$, our goal is to find scoring functions $f_1, \cdots, f_M \in \mathcal{H}$ for all reviewers, such that the distance between the true qualities $\mathbf{x}$ and qualities recovered by these selected scoring

---

[2]Overfitting here is in the similar sense to degenerated cases such as setting $k = n$ in k-means clustering of $n$ samples, or in low-rank matrix completion where $k$ is the target rank, $n$ is the matrix size.

[3]A scoring function $f$ is *convex* if for any $x_1, x_2 \in \mathbb{R}$ and for any $t \in (0, 1)$, we have $f(tx_1 + (1 - t)x_2) \leq tf(x_1) + (1 - t)f(x_2)$. Function $f$ is concave if $-f$ is convex.

functions is minimized. In this paper, we use the Euclidean distance. This leads to the following general optimization-based framework, which we term *Least Square Calibration (LSC)*. Note that in general LSC is a *functional optimization problem (FOP)* with: (1) functional variables $\mathbf{f} = \{f_j \in \mathcal{H}\}_{j \in [M]}$; (2) item quality variables $\mathbf{x} = \{x(i)\}_{i \in [N]}$; (3) noise variables $\epsilon = \{\epsilon_j^\ell\}_{j \in [M], \ell \in [|I_j|]}$.

---

**Least Square Calibration (LSC)**

$$\min_{\mathbf{x}, \mathbf{f}, \epsilon} \quad \sum_{j=1}^{M} \sum_{\ell=1}^{|I_j|} (\epsilon_j^\ell)^2 \tag{2}$$

$$\text{s.t.} \quad y_j^\ell = f_j \left( x(I_j^\ell) + \epsilon_j^\ell \right) \qquad \forall j \in [M], \ell \leq |I_j|$$

---

where the set cardinality $|I_j|$ is the number of items reviewer $j$ reviews. LSC tries to find the scoring functions and true qualities that best fit all the review scores in terms of the least square error.

**The rationale behind LSC** can be seen from several perspectives. First, fix any (unknown) parameters $\mathbf{x}$ and $\mathbf{f}$, define random variable $Y_j^\ell = f_j \left( x(I_j^\ell) + \epsilon_j^\ell \right)$ with randomness inherited from the noise $\epsilon_j^\ell$. Consider the following parameter estimation problem[4] — given observed realizations of $Y_j^\ell$ as $y_j^\ell$, find parameters $\mathbf{x}$ and $\mathbf{f}$ to maximize the likelihood of the observation. It is not difficult to see that the optimal solution to LSC is the maximum likelihood estimator, when $\epsilon_j^\ell$ is i.i.d. Gaussian noise with mean $0$. Second, it turns out that LSC can also be viewed as a variant of the classic *matrix seriation* problem. In Section 4, we will discuss this connection in more details as well as the connection of LSC to linear regression.

## 3.1 Linear Scoring Functions with No Noise

LSC is a general calibration framework for recovering the true qualities of the items. One cannot hope to solve it efficiently without any assumption on the scoring function hypothesis class, since functional optimization problems are generally intractable. Fortunately, we will show that for a broad class of commonly used hypothesis classes, LSC can be formulated as convex programs and therefore be solved efficiently. In some situations, we can even show that LSC can provably recover the correct item qualities perfectly under mild assumptions.

As a warm-up, we first restrict our attention to perhaps the mostly widely studied hypothesis class, i.e., linear functions. We start with the simple situation with no noise. In this case, it is not difficult to show that the following linear feasibility problem[5] solves the LSC Problem (2). We omit the proof of this claim since a more general result will be proved in subsection 3.2.

---

**LSC with linear scoring functions and no noise**

$$\min_{\mathbf{x}} \quad 0 \tag{3}$$

$$\text{s.t.} \quad x(I_j^\ell) - x(I_j^{\ell-1}) \geq 1 \qquad \forall j \in [M], \forall 2 \leq \ell \leq |I_j|$$

$$\frac{x(I_j^\ell) - x(I_j^{\ell-1})}{y_j^\ell - y_j^{\ell-1}} = \frac{x(I_j^{\ell+1}) - x(I_j^\ell)}{y_j^{\ell+1} - y_j^\ell} \qquad \forall j \in [M], 2 \leq \ell \leq |I_j| - 1$$

---

With the prior knowledge that the scoring functions are all linear functions and that there is no noise, it is easy to see that the true item qualities $\mathbf{x}^*$ and scoring functions $\mathbf{f}^*$ must be a feasible solution to LP (3). However, we cannot guarantee that LP (3) does not have any other feasible solutions whose induced item order disagrees with $\mathbf{x}^*$. This could happen, for example, when the reviewers do not review enough items, or their assigned items do not have enough overlap to perform a full calibration. Therefore, it is an important question to understand under what condition we can (essentially) recover all item qualities using LP (3), in the following sense:

---

[4]The parameters in this estimation problem includes functional parameters.

[5]Here to guarantee $x(I_j^\ell)$ to be strictly larger than $x(I_j^{\ell-1})$ and we w.l.o.g. use $\geq 1$ for convenience since rescaling the solution will not change ranking.

**Definition 3.1** (Perfect Recovery). *A set of qualities* $\{x(i)\}_{i \in [N]}$ *is said to* perfectly recover *the true qualities* $\{x^*(i)\}_{i \in [N]}$ *if there exists* $a > 0$ *and* $b$ *such that* $x(i) = ax^*(i) + b$ *for every* $i \in [N]$.

We remark that perfect recovery is a stronger condition than recovering just the *correct order* of the items. It additionally requires that the quality gaps among different items are proportional to the gaps of true qualities. One might wonder whether such a strong goal of perfect recovery may be too good to be possible. Surprisingly, our next result formally show that if the reviewer assignments satisfy some reasonable conditions, it is indeed possible to hope for perfect recovery. To introduce our result, we introduce a novel notion of a *review graph* and the property of *recover-resilience*.

**Definition 3.2** (Review Graph and Recovery-Resilience). *A review graph is an undirected multi-graph[6]* $G = (V, E)$ *where* $V$ *is the set of reviewers, and there are* $|I_i \cap I_j|$ *edges connecting every two reviewers* $i, j \in V$. *A review graph* $G$ *is said to be* recovery-resilient *if for any item assignments inducing* $G$ *and any given review scores* $\{y_j^l\}_{j,l}$, *any solution to LP* (3) *is a perfect recovery.*

That is, the review graph is a graph that connects reviewers to each other according to the number of common items that they reviewed. It turns out that the recovery-resilience of a review graph is fully characterized by a novel notion of *graph connectivity*, which we coin the *double connectivity*.

Double connectivity naturally generalizes the standard notion of graph connectivity. Recall that a connected component of the reviewer graph is simply a subset of reviewers $S \subseteq [M]$ who connects to each other. We will refer to the items they have collectively reviewed, $C(S) = \bigcup_{i \in S} I_i$, as the items *covered* by component $S$. A connected component of a graph can be identified through a simple `Repeat-Union` procedure, where we start by treating each node as a component and then repeatedly joining two components whenever they share at least *one* edge. Our notion of *doubly-connected component* include all components that can be found using the same joining procedure, except that each time we join two components only when they share at least *two* edges. A formal procedure, coined `Repeat-Union2`, is given in Algorithm 1 below.

---

**Algorithm 1** `Repeat-Union2`

---

**input** Review graph with $N$ item, $M$ reviewer, and item assignment set $\{I_j\}_{j \in [M]}$
1: $U \leftarrow \{\{1\}, \{2\}, \ldots, \{M\}\}$
2: **while** $\exists S_1, S_2 \in U$, s.t. $|C(S_1) \cap C(S_2)| \geq 2$ **do**
3: $\quad U \leftarrow U - S_1 - S_2 + (S_1 \cup S_2)$ (i.e., merge $S_1, S_2$)
4: **end while**
5: **return** $U$

---

**Definition 3.3** (Doubly-connected component). *Any component that can be generated by Algorithm 1 is called a* doubly-connected component.

The following theorem shows that to guarantee perfectly recovered qualities from LP (3) under linear scoring functions without noise, all we need is a *doubly-connected* component that covers all items.

**Theorem 3.4.** *A review graph* $G$ *is recovery-resilient if and only if the review graph* $G$ *has a doubly-connected component* $S$ *that covers all items, i.e.* $C(S) = [N]$.

*Proof Sketch.* The proof is somewhat involved particularly for the necessity of double-connectivity. We defer the full proof to Appendix A.1 and only describe the proof sketch here. For the "if" direction, the key idea is to use scores of at least two shared items between two reviewers $j_1$ and $j_2$ to calibrate the other items they reviewed in $I_{j_1}, I_{j_2}$. After this calibration, we can effectively treat these two reviewers as a single "reviewer" (formally, a doubly-connected component) who reviews all the items $I_{j_1} \cup I_{j_2}$. Algorithm 1 follows exactly this inductive procedure by repeatedly merging two components into a single component, until there is a "reviewer" that reviewed the entire item set $[N]$, and these are the calibrated qualities that we want.

The "only if" direction is more challenging. Here, we have to prove that for *any* reviewer graph that does not have a doubly-connected component covering $N$, there must exist an instance where perfect recovery is impossible. The main difficulty lies at crafting such an instance, by specifying paper assignment, true item qualities etc., for any such graph. Our proof leverages nice properties of prime

---

[6]A multi-graph is a graph that allows multiple edges between two vertices.

numbers to carefully construct the instance such that it will provably lead to at least two linearly independent solutions of LP (3). □

**Remark 3.5.** *Theorem 3.4 provides useful and practical insights for reviewer assignments in real applications. Specifically, it appears that stronger connectivity among reviewers during review assignment, defined formally in the sense of Definition 3.3 could be more helpful to calibrate their reviews. In practice, such connectivity can be achieved as follows: (1) run Algorithm 1; (2) if the algorithm stops without a doubly connected component covering all items, then simply ask any reviewers in one component to generate two reviews for two additional items covered by the other component so that these two components now become double-connected. This procedure is actually common during today's academic conference review as a way for calibration during post-rebuttal discussions. Interestingly, Theorem 3.4 serves as a theoretic justification for this practice. In Appendix B.2, we provide experiment results which further suggest that the topological structure of double connectivity exhibits stronger robustness against perception noise.*

## 3.2 Linear Scoring Functions with Noise

The noiseless case discussed in the previous section has an important feature: each reviewer's reviewing scores are monotone in the items' true qualities. This is a very useful property that could help the design of efficient calibration algorithms. However, it is also a restricted assumption that is often violated in practice. In real-world conference reviews, it is very common to see two reviewers reviewing the same pair of items but rating them in different orders. This is typically explained by reviewers' noisy perception of the paper qualities [10, 21, 5, 16, 27]. This leads to our model of perception noise in score generation procedure (1).

---

**LSC with linear scoring functions and with noise**

$$\min_{\mathbf{x},\epsilon} \quad \sum_{j=1}^{M} \sum_{l \leq |I_j|} (\epsilon_j^\ell)^2 \tag{4}$$

$$\text{s.t.} \quad \widetilde{x}_j^\ell - \widetilde{x}_j^{\ell-1} \geq \frac{y_j^\ell - y_j^{\ell-1}}{C} \qquad\qquad \forall j \in [M], 2 \leq \ell \leq |I_j|$$

$$\frac{\widetilde{x}_j^\ell - \widetilde{x}_j^{\ell-1}}{y_j^\ell - y_j^{\ell-1}} = \frac{\widetilde{x}_j^{\ell+1} - \widetilde{x}_j^\ell}{y_j^{\ell+1} - y_j^\ell} \qquad\qquad \forall j \in [M], 2 \leq \ell \leq |I_j| - 1$$

$$\widetilde{x}_i^\ell = x_i^\ell + \epsilon_i^\ell \qquad\qquad \forall j \in [M], 1 \leq \ell \leq |I_j|$$

---

As discussed before, we assume each $\epsilon_j^\ell$ is an independent noise added to the true quality of the item. Let $\mathcal{H}_L(C)$ denote the set of all increasing linear scoring functions whose derivatives (whenever exists) are upper bounded by some constant $C$. More formally, $C = \sup\{\frac{f(x_i)-f(x_j)}{x_i-x_j}|\forall x_i, x_j\}$. (In practice, $C$ can be empirically chosen to be a large constant if prior knowledge is lacking.) After adding the noise to the linear constraints in LP (3), our LCS with linear scoring functions becomes the above convex program, whose validity is guaranteed by the following theorem.

**Theorem 3.6.** *Convex Program (4) is equivalent to LSC with $\mathcal{H} = \mathcal{H}_L(C)$ in the following sense: for any optimal solution $(\mathbf{x}^*, \epsilon^*)$ to Program (4), there exists $\mathbf{f}^* = \{f_j^* \in \mathcal{H}_L(C)\}_{j \in [M]}$ such that $(\mathbf{x}^*, \epsilon^*, \mathbf{f}^*)$ is optimal to LSC Program (2).*

Theorem 3.6 shows that we can recover the same item order by solving $\mathbf{x}$ from LSC or by solving CP (4). The proof of Theorem 3.6 relies on a general procedure to argue the equivalence between a functional optimization problem and a variable optimization problem. We defer the formal arguments to Appendix A.2.

## 3.3 Handling Other Classes of Scoring Functions

It turns out that the above approach in Section 3.2 can be generalized to handle prior knowledge on various types of function shapes such as linearity, concavity and convexity, reference information (e.g., knowing the true quality of a few items), derivative information such as Lipschitz continuity, etc.

For example, suppose all we know about the scoring functions is that they are monotone increasing, then Program (4) can be adapted to this case simply by removing its second linearity constraint and only using the ordinal constraint $x_j^\ell - \widetilde{x}_j^{\ell-1} \geq \frac{y_j^\ell - y_j^{\ell-1}}{C}$ with a sufficiently large constant $C$. As another example, if reviewers are known to have convex scoring functions, we can change the linearity constraint to the convexity constraint $\frac{\widetilde{x}_j^\ell - \widetilde{x}_j^{\ell-1}}{y_j^\ell - y_j^{\ell-1}} \leq \frac{\widetilde{x}_j^{\ell+1} - \widetilde{x}_j^\ell}{y_j^{\ell+1} - y_j^\ell}$. These can all lead to similar guarantees as Theorem 3.6.

Additionally, another advantage of our optimization LSC framework is that it can handle a *mixed* set of reviewer scoring functions, *if such more refined prior knowledge is available*. Specifically, in many applications, the algorithm designer may have different amount of prior knowledge about different reviewers since some (senior) reviewers have been in the system for a long time whereas some just entered the system. Consequently, the designer may know, e.g., some reviewers' scoring functions are linear, some are convex or concave, whereas for some other reviewers, the designer knows nothing beyond monotonicity. Our LSC framework can be easily adapted to handle such mixed prior knowledge since its constraints can be "personalized" for each reviewer. Certainly, more specific or detailed prior knowledge about the hypothesis class will lead to better calibration, as also shown in our experiments. However, we remark that what hypothesis class is more realistic or justifiable will likely depend on concrete applications on hand, and is beyond the scope of this paper — our focus here is on the generic methodologies. We will demonstrate this kind of modeling strength of the LSC framework in the experiments section.

## 4 Connection to Other Problems

Our LSC framework shares connections to several other problems in statistics and machine learning. In the hope of providing further intuition and justification for our approach, in this section we elaborate on these connections.

### 4.1 Linear Regression

Linear regression [23] is a classic approach to modeling the relationship between two or more variables as a linear function. *Ordinary least square (OLS)* is a method to fit such a linear model by minimizing the sum of squared difference between the observed dependent variable and the predicted dependent variable. More specifically, OLS can be expressed by the formulation in Program (5).

$$\min_{\alpha,\beta,\epsilon} \quad \sum_{j=1}^{M} (\epsilon^\ell)^2 \qquad (5)$$
$$\text{s.t.} \quad y^\ell = \alpha \cdot x^\ell + \beta + \epsilon^\ell \quad \forall l$$

Linear regression formulation

$$\min_{\mathbf{x},\alpha,\beta,\epsilon} \quad \sum_{j=1}^{M} \sum_{\ell=1}^{|I_j|} (\epsilon_j^\ell)^2 \qquad (6)$$
$$\text{s.t.} \quad y_j^\ell = \alpha_j \cdot (x(I_j^\ell) + \epsilon_j^\ell) + \beta_j \quad \forall j \in [M], l \leq |I_j|$$

LSC formulation

Observe that the formulation of linear regression is structurally similar to our LSC (2) when $\mathcal{H} = \mathcal{H}_L$, which is formulated in Program (6) above. There are two major differences between our LSC in Program (6) and OLS in Program (5). First, the true item qualities $\mathbf{x}$ in OLS (5) is known, whereas in our LSC (6), $\mathbf{x}$ is unknown and treated as variables to optimize. Second, with the additional challenge of not knowing $\mathbf{x}$, the LSC also has an advantage — we have multiple reviewers (indexed by $j$) with multiple linear functions/models who will generate scores for the same $\mathbf{x}$. These make LSC relevant to but quite different from OLS linear regression.

### 4.2 Matrix Seriation Problem

Our framework is also closely relevant to a well-known difficult combinatorial optimization problem called *matrix seriation* [12, 15]. The problem looks to find a consistent ordering of the columns of a matrix. More specifically, given a set of $n$ objects and a (partially observed) data matrix $A$ in which each row of $A$ reflects partial observations of these objects, the goal of seriation is to find a linear ordering of these $n$ objects that best fit the observed data according to certain loss function.

The matrix seriation problem has its roots in many disciplines, with applications such as sparse matrix reordering [4], DNA sequencing [6], and archeological dating [7]. There are many variants of the matrix seriation problem. Below we define a popular version with the $\ell_2$ loss function.

**Definition 4.1** ($\ell_2$-Matrix Seriation). *Given a partially observed matrix $A \in \mathbb{R}^{m \times n}$ in which each entry $A_{ij} \in \mathbb{R}$ or $A_{ij} = *$ (not observed), find matrix $B \in \mathbb{R}^{m \times n}$ that minimizes $\|A - B\|_2^2 = \sum_{(i,j):A_{ij} \neq *} (A_{ij} - B_{ij})^2$ subject to $\forall i, j \in [m], q, p \in [n], B_{i,q} \leq B_{i,p} \iff B_{j,q} \leq B_{j,p}$. That is, in matrix $B$, the ordering of each entry according to the column is consistent across each row.*

Review score calibration can be naturally modeled as a matrix seriation problem. We have a set of items, and a data matrix $A$ in which each row represents a reviewer's rating scores to (only) his/her assigned items. Our goal is to find an ordering of the items that is most consistent with the scoring matrix $A$ under the assumption that each review has a *monotone* scoring function. In the following, we show that the seriation problem is a small variant of our LSC problem.

**Theorem 4.2.** *The $\ell_2$ Matrix Seriation (4.1) problem can be solved by the following Functional Optimization Problem ($\mathcal{H}_{mono}$ contains all monotone increasing functions):*

$$\min_{\mathbf{x},\mathbf{f}} \quad \sum_{j=1}^{M} \sum_{\ell=1}^{|I_j|} (\epsilon_j^\ell)^2 \tag{7}$$

$$\text{s.t.} \quad y_j^\ell = f_j(x(I_j^\ell)) + \epsilon_j^\ell \ \text{ and } \ f_j \in \mathcal{H}_{mono}, \qquad \forall j \in [M], \ell \leq |I_j|$$

It is worthwhile to compare FOP (7) with our LSC (2) with monotone scoring functions as they have very similar formulations. The difference is that in FOP (7), the noise term $\epsilon_j^\ell$ is added after applying the scoring function, whereas in LSC (2) it is added to $x(I_j^\ell)$ inside the function. Notably, this seemingly small difference turns out to greatly affect the tractability of the problem. Indeed, the matrix seriation problem is known to be computationally *intractable* [8]. In contrast, our LSC (2) with monotone scoring functions can be efficiently solved via convex programming due to the noise inside scoring functions (as assumed by most previous works in this space [10, 21, 5, 16, 27]).

## 5 Experiments

A major challenge when evaluating the performance of calibration algorithms is that their performance measures rely on the underlying true item qualities. Unfortunately this information is unattainable in most applications — indeed, if we already know the true qualities, peer reviews would not be needed any more. Besides, due to anonymity of reviewers in all public peer review data that we are able to find, the underlying network of review graph cannot be recovered. Therefore, like many other works in this domain [16, 26, 27], we evaluate our algorithms primarily based on synthetic data where we do know the true qualities. To be more realistic, the distribution parameters of our synthetic data are chosen based on ICLR 2019 review scores [1]. Our experiments are designed according to the review procedure of academic papers, we thus also refer to items as *papers* in this section. In addition, we include an experiment on a real-world dataset [22] in Section 5.2. While our model still consistently demonstrates good performance, we still want to raise a caveat here that strategic manipulation could be a non-negligible factor in the process of peer grading [29, 24]. Given that such strategic behavior is rarely observed in peer review setting such as academic conferences, it therefore remains a crucial task to obtain a real-world dataset for future work of this domain.

### 5.1 Evaluation on Synthesized Dataset

**Dataset Generation** Each paper's true quality is drawn from Gaussian distribution $\mathcal{N}(5.32, 1.2)$ truncated to be within $[0, 10]$,[7] where the mean $5.32$ and standard deviation $1.2$ is estimated based on the distribution of ICRL2019 review scores. Our random assignment algorithm ensures that most papers will get reviewed by roughly the same number of reviewers. Each reviewer is randomly assigned a scoring function from pre-specified family. Before applying the scoring function, a zero-mean i.i.d. Gaussian noise $\epsilon \sim \mathcal{N}(0, \sigma)$ is added to the true quality of each paper for each reviewer. More details about the scoring function generate as well as the assignment algorithm can be found in Appendix B.

---

[7]Truncation is for comparison convenience. Untruncated scores will only be easier since boundary papers are easier to distinguish.

Table 1: Performance comparison under **linear scoring** setting (LSC is our method). The left and right side of the table respectively corresponds to the **noiseless** ($\sigma = 0$) and **noisy** ($\sigma = 0.5$) setting.

| Metric
Model | Pre. (%) | Avg. Gap | Pre. (%) | Avg. Gap |
|---|---|---|---|---|
| Average | $40.0 \pm 4.0$ | $0.78 \pm 0.08$ | $39.2 \pm 4.5$ | $0.80 \pm 0.08$ |
| QP | $97.1 \pm 1.7$ | $0.01 \pm 0.01$ | $69.2 \pm 4.6$ | $0.24 \pm 0.09$ |
| Bayesian | $76.2 \pm 4.0$ | $0.12 \pm 0.02$ | $71.5 \pm 3.1$ | $0.17 \pm 0.03$ |
| LSC (mono) | $91.7 \pm 1.8$ | $0.02 \pm 0.01$ | $75.9 \pm 2.6$ | $0.12 \pm 0.02$ |
| LSC (linear) | $\mathbf{100 \pm 0}$ | $\mathbf{0 \pm 0}$ | $\mathbf{80.1 \pm 2.9}$ | $\mathbf{0.08 \pm 0.01}$ |

**Evaluation Metrics** We adopt two different metrics to quantitively and comprehensively evaluate the performance of our models. Let $T \subset [N]$ be the ranked list of $n$ papers with highest true qualities and $S \subset [N]$ denote the ranked list of $n$ papers selected by any algorithm. In all our experiments, we set $|S| = |T| = 0.1N$, i.e., an acceptance ratio of $10\%$ in our simulated conference setting.

- **Precision** $\rho(S, T) = \frac{1}{|S|} \sum_{i \in S} \mathbf{1}[i \in T]$ measures the ratio of the top papers in $T$ that are selected into set $S$. The higher precision is, the more papers with truly high qualities are accepted.
- **Average Gap** $\sigma(S, T) = \frac{1}{|T|} \sum_{i \in T} x^*(i) - \frac{1}{|S|} \sum_{i \in S} x^*(i)$ measures the gap between the average true qualities of the best papers in $T$ and average true qualities of the papers in $S$. It characterize how far the recovered average paper quality is from the optimal. The smaller this gap is, the better.

**Baselines** We consider three competitive baselines. The first is a widely used heuristics which simply uses the averaged reviewers' scores as a prediction of the paper's true quality, and accordingly select the best papers. We will refer this model as the **Average**. The second baseline is based on a *quadratic program* (**QP**) proposed by Roos et al. [21]. The third baseline proposed by Ge et al. [10] uses a Bayesian model (**Bayesian**) to calibrate paper scores. The latter two baselines both assume linear reviewer scoring functions with noise that is similar to our modeling assumption.

**Experiment Setup** We first test the situation with linear scoring functions in order to have fair comparisons with previous methods which mostly assume linearity. Parameters are set as $N = 1000, M = 1000, k = 5$. We compare two settings: (1) $\sigma = 0$ (*noiseless* case); (2) $\sigma = 0.5$ (*noisy* case). All reported results are averaged over 20 trials. All of our models are implemented with the Gurobi Optimizer [11][8]. All our algorithms can be solved very efficiently in seconds, we thus will not present running time results.

**Results** Table 1 presents the experiment results that compare our methods with the three baselines. Notably, our model consistently outperform all baselines under both metrics in both settings. In the noiseless setting with linear scoring functions, the experiment results confirm our theoretical analysis, as we indeed observe the perfect recovery by our model under this typical conference setup. In addition, while both LSC (linear), QP and Bayesian specifically models the prior knowledge of linear scoring functions, our model demonstrates the more robust performance regardless of the noisy environment in both metrics. This suggests that our model meets the objectives that we want to accept the best papers and maximize the overall qualities of the selected papers. We also investigate how the model performance changes according to the hardness of the problem instance, such as the effect of changing number of papers per reviewer $k$, the paper to reviewer ratio $N : M$, and the noise scale $\sigma$ in our empirical study and plot the results in figures. In all experiments, LSC still consistently outperforms baselines, and more details can be found in the Appendix B.

## 5.2 Evaluations on the Peer-Grading Dataset

To test our model beyond synthesized dataset, we use a well-known public Peer-Grading dataset from [22] that collects the peer grading scores of questions in 6 different homework submissions in an Algorithm & Data Structures class at the University of Hamburg. We use the average score graded by the TAs as the ground truth quality $x^*$ of each paper (homework submission). In Table 2, we list

---

[8]The source code can be found at `https://github.com/lab-sigma/lsc`

Table 2: Performance comparisons in **Peer-Grading Dataset**.

| Metric \ Model | Average | QP | Bayesian | LSC (mono) | LSC (linear) |
|---|---|---|---|---|---|
| Pre. (%) | 80.9 | 80.3 | 79.4 | 78.5 | **82.2** |
| Avg. Gap | 0.48 | 0.40 | 0.47 | 0.55 | **0.38** |

the precision of the LSC models and other baseline on selecting the top 50% of the first homework submission. Our model has the best calibration result. Moreover, the LSC model shows an even more clear edge in ranking-based metrics, meaning the order of the recovered quality matches better with the ground-truth. We defer the details of these results to Appendix B.6.

### 5.3 Mixed Set of Scoring Functions and Robustness to Mis-Specified Prior Knowledge

To study the robustness of the different methods in scenarios where the scoring function assumption no longer holds or is mis-specified, we consider settings where some percentage of reviewers' scoring functions are instead *arbitrary random* (non-linear) monotone functions. Figure 1 plots the precision curve of our methods and baselines with varying percentage of linear scoring functions. It shows that 1) our LSC (linear) consistently outperforms prior work (QP, Bayesian) that also assumes linear scoring functions. 2) As the percentage of linear function decreases, the gap between LSC (linear) and LSC (mono) becomes closer, and when all functions are random monotone functions, LSC (linear) is $4\%$ behind LSC (mono) due to its mis-specified prior knowledge in the noiseless setting. 3) LSC (mix) — which has accurate prior knowledge about which reviewers are linear — significantly outperforms LSC (linear) and all other models, showing that good prior knowledge is indeed quite helpful for calibration.

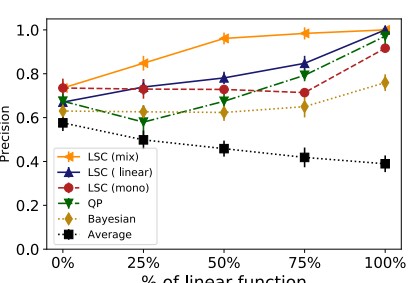

Figure 1: Performance comparisons in the noiseless and mixed setups with linear scoring functions and arbitrary monotone functions. Only LSC (mix) has prior knowledge of every reviewer's scoring function type.

Appendix B.4 contains additional experiments for noisy scores and mixed settings with monotone, convex and concave functions. The results further demonstrate the usefulness of prior knowledge for calibration and thus also justified the value of having a flexible calibration methods like LSC.

## 6 Conclusion

This paper presents a simple yet powerful framework for calibration in peer review systems, which exploits both the robustness of linear regression methods and the topological structure of review graphs. Moreover, our empirical and theoretical results provide a general guideline on the assignment rules in peer review for more effective calibration. In follow-up work, we hope to build upon our flexible framework for the extension to a wider spectrum of scoring function hypothesis classes, as well as a broader family of peer review systems beyond the classical format of academic conference. For example, AAAI-2021 recently started to adopt a review process[9], where in the first phase senior reviewers review all the papers and make summary rejections. The remaining papers will go to the second phase and receive additional reviews for the final decisions. It remains interesting open problems to develop novel calibration methods for such two-phase peer review process, or alternatively, to design the peer review mechanism to work better with calibration methods.

---

[9] https://aaai.org/Conferences/AAAI-21/reviewing-process/

# 7 Acknowledgements

We thank all the anonymous reviewers for their helpful comments, especially the discussions on real-world datasets. This research is supported by a Google Faculty Research Award and the Ministry of Education, Singapore, under its Academic Research Fund Tier 2 (MOE2019-T2-1-045).

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
