# OpenReview forum: "Least Square Calibration for Peer Reviews"
_NeurIPS.cc/2021/Conference — NeurIPS 2021 Poster_

### Official Review · Reviewer_Emh1 · 2021-07-15

**Rating:** 6
**Confidence:** 2

**Summary:**

The paper addresses the problem of review calibration, i.e., to
estimate true paper qualities from reviewer ratings, if they
are monotonic functions in the true quantities. The authors model
the problem as an optimization problem that aims to minimize
input noise on the reviewer functions. The authors derive explicit
linear programs for the case of linear reviewer functions with
and without noise and sketch further variants. They discuss differences
of their approach to linear regression and matrix seriation.
In an experiment with synthetic data (and thus given ground truth),
they compare their method against two baselines from the literature
and show improvements both in the case of linear reviewer functions
being the groundtruth as also for more general reviewer functions.


**Limitations And Societal Impact:**

yes

**Main Review:**

The paper is very well written and a joy to read. The proposed method
is plausible and the experimental results look very good, though only
synthetic data has been used.

I have only three minor points with the paper:
1. The proposed method is a pure optimization solution, thus the
    paper maybe will be better positioned in an optimization conference?
2. It might be practically interesting only for a small, special audience (PC chairs).
3. If there is no way to observe the groundtruth as the authors allude to
   ("if we already know the true qualities, peer reviews would not be needed
   anymore"), this would raise the question what exactly these "true qualities"
   then are. Maybe the field of peer review calibration needs to find
   some way to make them observable.


**Time Spent Reviewing:**

2.5

---

> ### Author Response · Authors · 2021-08-10
>
> We thank the reviewer for the encouraging feedback and for appreciating our results. Below we briefly respond to the reviewer's three comments:
>
> 1. The calibration problem in peer review is an unsupervised learning problem (just like the classical machine learning problem of *clustering*), so we believe it is well suited for NeurIPS.
>
> 2. We remark that our model is applicable to many crowdsourcing tasks which employ crowdsourced workers to determine ranking or winners. We use peer review just for illustrative purpose.
>
> 3. We fully echo this point and believe that this is itself a very intriguing open problem for peer review calibration (see more discussion in our global response), though it is out of the scope of our current paper.

---

### Official Review · Reviewer_o3ak · 2021-07-16

**Rating:** 7
**Confidence:** 3

**Summary:**

This paper studies calibration in the peer review process. Unlike most previous work, in this paper, the structure of reviewer miscalibration needs to only be monotone (in the true score of the paper being reviewed).  The authors also study the cases where: miscalibration is linear, and there is added noise.  The main contribution is framing the calibration problem as a (specific) functional optimization problem. Theoretically, they prove that optimal solutions to their optimization problems yield the correct underlying ordering of the papers (by true score). In comparison to 3 baseline algorithms the proposed approach tends to perform best in simulated experiments.

**Ethical Concerns:**

This paper does not include a statement related to ethics but I believe it is worth thinking about/discussing. Applying this framework could be problematic if discovering incorrect calibration functions, and leading to the rejection of papers that would otherwise be accepted. It might be worth it to analyze the number of flipped accept/reject decisions when compared to average vs. true score--or some similar analysis. In a similar spirit, it’s worth it to think through and describe when a naive application of the proposed approach would fail and what the consequences would be.

**Limitations And Societal Impact:**

There are a handful of limitations with this work.  First and foremost, the monotonicity requirements on the score generation. While this is a significant limitation, a) this could be reasonable in practice and b) virtually all related work makes the same assumption.

Another limitation is the assumptions required on the type of scoring functions (linear/monotonic).  This seems to be difficult to judge a-priori and important: e.g., the QP approach outperforms the proposed LSC-monotone approach in the noiseless case.  That being said, the noiseless case is perhaps a bad approximation of reality.

As mentioned above, the expectation of doubly connectedness in the review graph is a limitation. Do all the results require this? Do all the simulations guarantee that the review graphs are doubly connected? And, if so, what happens if the review graphs break this assumption?

Finally, I’d like to better understand the design choice of applying noise to the papers rather than the reviewers.

**Main Review:**

*Originality*:
Framing the calibration problem as functional optimization seems novel to me and is interesting.  In experiments it also tends to yield superior results (i.e., precision/GAP) than other competing methods. Similarly, while subtle, I think the idea of putting the noise term _inside_ the score generation (equation 1), is quite useful and makes the problem easier to solve (than putting the noise term outside). I don’t see a problem with this approach immediately; assuming it is not problematic, this could be used in further research on calibration.

Additionally, I found remark 3.5 especially interesting. I was unaware that additional reviews are requested in order to achieve better calibration (I thought additional reviews were requested for other reasons).  Authors: can you give a citation or anecdotal evidence of this? Either way, the remark implies a strategy of holding out some reviewers and assigning them after an initial set of reviews for the sake of better calibrating the reviewer scores.


*Quality*:
The work seems technically sound: the “if” direction of Theorem 3.4 seems correct; the “only if” direction I’m a bit confused about. Theorem 3.6 also seems correct.  All functional optimization problems outlined seem correct as well.

In the experimental setup: M=1000 and N=1000 seems unrealistic. Why not use the exact parameters from ICLR? I’m curious about whether/how these parameters affect the results.  Moreover, I’m interested in how the distribution of paper (true) scores affect the results (in terms of precision/gap). I’m also curious about other evaluation metrics: how many papers would be wrongly accepted/rejected when using each of the baselines? The precision/gap get at this, but it would be illuminating to calculate statistics like this for each method.

Finally, in the experimental section, it would be useful to describe why the proposed method outperforms the other baselines in each case, especially in the case of linear functions (with/without noise).

*Clarity*:
Overall, the paper was clear. However, some details related to clarity could be improved. For example: what is the range of true scores, $x*$? Of the scores x in the LP?  In the LSC function with linear scores and no noise: it was at first confusing to me that the scoring functions do not appear; consider adding them or explaining this in the text. In definition 3.1: I’m a bit confused if $x(i)$ is a linear function of the true score and $y(i)$ is a linear function of $x(i)$.  Also, can you explain why the first constraint in CP 4 is necessary? Why does Theorem 3.6 only imply that the true item ordering is recoverable but not the scores?

At first glance, Section 4 seems a bit superfluous.  That being said, I found 4.1 helpful. Perhaps some of this information could be condensed and added into previous sections. This would give room for more experimental results.

In the experiments section, k is referenced before it is defined.

In Figure 1: why does the precision of Average go down with increasing fraction of linear functions?  Moreover, it seems like EQ performs much worse when noise is added; can you explain why?

There are grammatical errors in the proof sketch of Theorem 3.4 (there are extra “the”s).

Algorithm box 1 may not be necessary (just describe it in words and put the algorithm in the appendix).


*Significance*:
I found the result from 3.4 interesting, however, I’m unsure of how technically novel it is. Moreover, I’m curious as to how often we’re likely to get a doubly-connected component that covers all papers in practice.  I think it would be worth it to check this using real data and standard (max-similarity) matching.

Applying function optimization to reviewer calibration is reasonable and I’d be very curious to see how this works in practice.  Similarly, I think it would be worthwhile for program chairs to experiment with this style of calibration in practice.


EDIT AFTER AUTHOR RESPONSE:  Thank you authors for clearing up some of my confusion.  Implementing the changes you identified in your response (e.g., writing changes, more realistic simulation and extrapolation of ethical concerns) will improve the paper.

**Time Spent Reviewing:**

5

---

> ### Author Response · Authors · 2021-08-10
>
> We sincerely appreciate the reviewer's detailed comments and encouraging feedback.
>
> Regarding the "citation or anecdotal evidence" for requesting more reviews to achieve better calibration, we   actually did not see this idea in any previous research of calibration and this idea comes out just as a natural consequence of our Theorem 3.4.  However, in reality, when area chairs request additional reviews for some paper, we would expect these reviews to help the AC to calibrate his/her set of papers.
>
> On the experiment setup, the reviewer made great suggestions to make our experiment results more realistic and illuminating. For your information, we actually did experiment with large parameters such as $M=5000, N=5000$, and observed that increasing the size of conference does not change any fundamentals about the performance of algorithm and baseline (and thus omitted).
> But we will make sure to include this empirical study on different conference size as well as the evaluation metrics you suggested in our final version.
>
> We appreciate the reviewer's valuable suggestions on our writing, and will properly implement them in our revision. Below is our response to your main points.
>
> - The reasoning behind the first constraint in CP 4 is discussed in Ln. 229-231. It is necessary for the numerical stability of our optimization program.
>
> - In our setting, it is only possible to recover the true item ordering but not the true quality, fundamentally because we can never recover the exact parameters of the linear scoring function. For example, observe that if we multiple 2 to all true qualities and divide the slopes of all true  linear scoring functions by 2, this would satisfy the optimization program just as the original solution.
>
> - To explain why the precision of Average go down with increasing fraction of linear functions, our understanding is that Average is a more robust heuristic when the reviewers' scoring functions are more likely to be random monotonic functions. We are not exactly sure about the "EQ" that you are referring to.
>
> - As you pointed out, we are also curious about how often we are likely to get a doubly-connected component that covers all papers in practice. Unfortunately, we are unable to obtain any real conference data, as they (public ones such as **ICLR**, **PeerRead**) enforces anonymity that forbids us to build the review graph.
>
> - We fully agree with your remark that it would be worthwhile for program chairs to experiment with this style of calibration in practice. We think this is an exciting future endeavor to pursue.
>
> We once again appreciate the objective reviews of our limitations and will correspondingly include more discussion in the final version. Here are some points we would like to clarify:
>
> - Note that in the Appendix B.2, we carefully designed the experiment to verify our Theorem 3.4. In the first two bar of Figure 2, we can directly observe how the recovery performance would change when the review graphs does not satisfy this doubly-connectedness condition.
>
> - Our model assumes noise in the perception of each paper review generated by each reviewer. We briefly discussed this design choice in Section 4.2, as it greatly affects the tractability of the problem.

---

> > ### Comment · Reviewer_o3ak · 2021-08-31
> > **Response/"EQ" --> "QP"**
> >
> > Authors,
> > Thank you your responses and clearing up most of my confusion.
> >
> > Also, apologies: by "EQ" I meant "QP" and was wondering about the degradation of performance in the noisy case.

---

> > > ### Author Response · Authors · 2021-08-31
> > >
> > > Thanks for your reply! We are glad that our response resolves most of your confusion. The reason that the performance of QP degrades more than other models under noisy setting is the following. QP uses maximum likelihood estimate to learn the parameters of reviewers’ linear scoring function with no regularization such as the order preserving constraint in LSC when we consider linear scoring situation. As the noise level increases, QP may even learn negative slope which could cause clear performance degradation.

---

### Official Review · Reviewer_zLc7 · 2021-07-18

**Rating:** 6
**Confidence:** 3

**Summary:**

The paper has proposed a new flexible framework, least square calibration (LSC), for selecting top candidates from peer ratings. The major contributions are summarized as follows:
a.	Proposed a model which could handle different types of miscalibration in peer reviews, and also incorporate various levels of prior knowledge
b.	Self-prove the effectiveness of LSC by both theoretically and empirically


**Ethical Concerns:**

None.

**Limitations And Societal Impact:**

None.

**Main Review:**

Originality
It is a good try to use LSC and other classes of linear scoring function for selecting top candidates for peer review, to the best of our knowledge, we do not see any previous papers have used this technique

Quality
The paper has provided rigorous mathematical proof for the LSC and other stuff, however, there are not many experiments done by the author, for example one more ablation experiment on how prior knowledge is incorporated and how that will impact the results. And this is actually facing a chicken-and-egg problem, if you know the quality, peer review is not needed anymore, but if you do not know the quality, you will not know how the performances will be in the true data. It might be also good to make experiments on real data. It outperforms all previous systems by a lot, which is really impressive. The paper has been too long (9 pages) and exceeded the maximum allowed length (8 pages).

Clarity
The paper is well-organized, used well-selected notations, and has provided rigorous proof of the method, but sometimes it is hard for the readers to follow.

Significance
The paper has proposed LSC to address miscalibration problem for peer review which outperforms all previous models.


**Time Spent Reviewing:**

4

---

> ### Author Response · Authors · 2021-08-10
>
> We thank the reviewer for the encouraging feedback and for appreciating our results. We will make sure to address your comments on our content, and improve the readability of our proofs in the final version.
> We fully echo your point that properly evaluate peer review algorithm in real world dataset has been a challenge faced by the entire community (see more discussion in our [global response](https://openreview.net/forum?id=rTxCRLXRtk9&noteId=JjSDCQWWtyz) on the top).
> Also note that due to the 9 page limit, we included many additional experiment results in the appendix (contained in the supplemental material).

---

### Official Review · Reviewer_PoXu · 2021-07-24

**Rating:** 5
**Confidence:** 2

**Summary:**

This paper tackles the problem of calibrating ratings, focusing on the application of scientific peer review. The idea is to normalize reviewers who interpret the rating scale differently. If one reviewer tends to use low ratings and another tends to use generous high ratings, the calibration step will equilibrate them. The calibration requires the two reviewers to have some papers in common, or at least both have papers in common with a chain of other reviewers. The idea is that each paper has an inherent quality, each reviewer perceives that quality with independent zero-mean noise, and each reviewer translates that perceived quality into a rating using a monotone function. Given this model of how ratings are formed, the authors propose an optimization problem to find the hidden qualities, function parameters, and error rates that minimize the squared errors that reviewers perceive. They call this Least Square Calibration (line 133). The authors show how to do this for linear functions with and without noise, and prove some conditions under which the exact qualities can be recovered, including an if and only if connection to doubly connected review graphs with no noise. The authors also show how to do run the optimization for non-linear functions. The authors discuss similarities with linear regression and matrix seriation. The authors provide simulations to show that their method works.

**Limitations And Societal Impact:**

yes

**Main Review:**

The authors address a real problem: how to calibrate ratings. The problem is hard because if one reviewer uses low ratings and another uses high ratings, it's hard to disentangle reviewer tendencies from paper qualities. If the papers the two reviewers rate are disjoint, it's especially hard, but not impossible, because the first and second reviewers may each have papers in common with a third reviewer or a chain of reviewers. The authors formulate the problem in a reasonable way. The solution is straightforward. I am not fully satisfied with the results for three reasons:

First, the authors don't discuss enough the problem of overfitting. If the function hypothesis class is too large, the optimization will almost certainly overfit. For example, in the definition of LSC at line 133, if the function f is a complex non-linear function with many parameters, the optimization will be able to find a solution with all the epsilon error terms zero, all the x values arbitrary, and the function f memorizing a mapping from x to y. When the function is linear, overfitting is somewhat controlled. When the function is non-linear, the authors will need some sort of regularization.

AFTER AUTHOR RESPONSE: the following second half of the previous sentence was not correct in my original review, as the authors' pointed out in their response: ", and the tests will need to be done in the usual ML way with the optimization being solved on different data than the tests are run."

Second, the empirical tests are simulations. For this application, testing on real data will be much more persuasive. Some peer review data is here: https://github.com/allenai/PeerRead/tree/master/data described in the paper https://arxiv.org/abs/1804.09635 . The data does include some numerical scores and includes accept/reject decisions that can be thought of as ground truth. There are other ratings data sets the authors can use. For example, data from peer grading systems should work, for example http://www.tml.cs.uni-tuebingen.de/team/luxburg/code_and_data/peer_grading_data_request_new.php . Here is another peer grading paper: https://www.cs.ubc.ca/~jrwright/mta/ . Data from review web sites might work too. Here is some review data: https://www.preflib.org/data/CD . There is no ground truth, so this may or may not work. One idea would be to use the average of all ratings as the ground truth and then try to recover the ground truth by calibrating a small subset of the users. A Mechanical Turk experiment would also be revealing and can be engineering to have a ground truth.

AFTER AUTHOR RESPONSE: I am impressed with the new empirical results. Thank you for doing that with amazing speed. I am glad that the review process led to a better paper with both synthetic and real results. Still, this is only one test and it is very hard to evaluate a new test produced during the review process.

Third, the empirical tests do not split the data into training set and test set, as far as I can tell. Perhaps I missed that. Without a proper split of training and testing, the authors cannot say whether their optimization overfit to the data.

AFTER AUTHOR RESPONSE: Thank you for your response and clearing up my misunderstanding. I looked back at the paper again at the main optimization problem on line 133. This IS an unsupervised problem. So, my third complaint above about no train-test split in the experiments is not valid, as the authors point out. Sorry about that.
However, I feel that my first argument is still valid. I don't agree that an unsupervised learning algorithm cannot overfit. For example, a clustering algorithm can get perfect results (by some metrics like purity or mutual information) by placing every data point in its own cluster: https://nlp.stanford.edu/IR-book/html/htmledition/evaluation-of-clustering-1.html This is a form of overfitting. Specifically, for this paper, in the definition of LSC at line 133, I don't see why some complex non-linear function f with many parameters couldn't be optimized to find a solution with all the epsilon error terms zero, all the x values arbitrary, and the function f memorizing a mapping from x to y. If the function class for f is too large (the analogy of setting k=n in k-means clustering), the optimization can memorize an answer that has all error terms zero.


I had one more question:

Is Definition 2.1 (Linear Scoring Function) only for the linear function without noise? What happened to the independent zero-mean noise? Is it zero? What is the linear scoring function with noise? Should it be a(x+epsilon)+b?

Here are some minor writing suggestions/corrections:

there exists cardinal scores estimators perform better than any ordinal ranking estimators
there exist cardinal score estimators that perform better than any ordinal ranking estimators

320: the same number of reviews
320: the same number of reviewers

assigned scoring function from pre-specified family
assigned a scoring function from a pre-specified family

bayesian model
Bayesian model

Reference [24] has no publication venue

**Time Spent Reviewing:**

4

---

> ### Author Response · Authors · 2021-08-10
>
> We thank the reviewer for the detailed comments, especially a curation of real world datasets. We will make sure to properly implement the writing suggestions you point out, and here we focus on your main concerns: (A) experiments with real-world data; (B) testing/training split and overfitting. Please see our [global response](https://openreview.net/forum?id=rTxCRLXRtk9&noteId=JjSDCQWWtyz) on the top for (A).
>
> Below we respectfully point out  that (B) is due to some misunderstanding   and  it should not have been a concern  in our setting. *Due to this critical misunderstanding, we therefore respectfully request the reviewer to re-evaluate the merit of our paper.*
> Note that  our problem here is an ***unsupervised*** learning problem, which does not have any issue with data splitting or overfitting. Specifically,  the data are the reviewers' ratings on papers. Our algorithm takes all these papers and their ratings as input, and output a ranking of the papers. This is similar to K-means clustering where the algorithm takes data points as input and cluster them into groups.  There is no "testing" phase in our setting. The algorithm's parameters are specific to the dataset so on a different dataset, parameters need to be reinitialized and retrained. This is just like how you would rerun K-means clustering on a different dataset. Therefore, there is no split between training and testing datasets. For the same reason, overfitting means that the algorithm overfits the training dataset so its performance degrades on testing dataset. Since there is no testing dataset in our setting, overfitting is not relevant to us.
>
> Next we clarify the minor question raised by the reviewer on Definition 2.1.
>
> Yes, this definition is for linear scoring function without noise because the scoring function only captures a reviewer's bias and scale, whereas noise is already captured in the score generation process defined in Equation (1). We apologize for the confusion. In the final version, we will use a better notation  $\tilde{x} (=x+\epsilon)$ in Def. 2.1 to distinguish it from true score $x$. Hope this would resolve the confusion.

---

> > ### Comment · Reviewer_PoXu · 2021-08-24
> > **thank you**
> >
> > Thank you for your response and clearing up my misunderstanding. I looked back at the paper again at the main optimization problem on line 133. This IS an unsupervised problem. So, my third complaint about no train-test split in the experiments is not valid, as you point out. Sorry about that. I will revise my review.
> >
> > However, I feel that my first argument is still valid. I don't agree that an unsupervised learning algorithm cannot overfit. For example, a clustering algorithm can get perfect results (by some metrics like purity or mutual information) by placing every data point in its own cluster: https://nlp.stanford.edu/IR-book/html/htmledition/evaluation-of-clustering-1.html This is a form of overfitting. Specifically, for this paper, in the definition of LSC at line 133, I don't see why some complex non-linear function f with many parameters couldn't be optimized to find a solution with all the epsilon error terms zero, all the x values arbitrary, and the function f memorizing a mapping from x to y. If the function class for f is too large (the analogy of setting k=n in k-means clustering), the optimization can memorize an answer that has all error terms zero. My apologies if I am missing something.
> >
> > I am impressed with the new empirical results. Thank you for doing that with amazing speed. I am glad that the review process led to a better paper with both synthetic and real results.

---

> > > ### Comment · Area_Chair_H7zv · 2021-08-24
> > > **Overfitting**
> > >
> > > Dear Reviewer "PoXu" (and authors),
> > >
> > > First, I agree with your point that choosing a very large class can result in fully fitting the data. Two points to this end on which I would love to know your thoughts:
> > >
> > > 1) Yes, k means with k=n can overfit.  But that does't mean k means is not useful, right? Over many years, the k means algorithm has been extensively studied and people have found heuristic and/or principled ways to choose k. In practice, one often manually chooses k via human insight and/or some heuristics, and moreover, many papers assume that k is somehow given or reasonably chosen by an oracle. A similar problem is low-rank matrix completion. Letting n denote the size of the matrix and k denote the target rank, choosing k=n will again overfit. The initial papers on this problem also assumed k as given. Wouldn't one consider this submission in a similar vein? Since this is an initial paper exploring the use of non-linear functions in miscalibration, it assumes the function class is exogenously chosen. Hopefully it will spur more research on heurstic/principled regularization/cross-validation/other methods to choose the function class.
> > >
> > > 2) Can the authors clarify the following: From eqn (2) it seems to me that if H is chosen to be very large, the the algorithm will essentially not do any correction. If this is correct, then this action is equivalent to what is done today (and so is fine). Is that right?
> > >
> > > What do you all think?
> > >
> > > Best,
> > > AC

---

> > > > ### Comment · Reviewer_PoXu · 2021-08-24
> > > > **Re: Overfitting**
> > > >
> > > > Dear AC and authors:
> > > >
> > > > The AC makes a good point in (1). Yes, this makes sense, although I think the paper should discuss the issue that regularization will be needed.
> > > >
> > > > I am also curious what the authors think about (2). It seems like if H is too large, the algorithm may find arbitrary corrections rather than no correction, but I could be wrong.
> > > >
> > > > Thanks! Great discussion.

---

> > > > > ### Author Response · Authors · 2021-08-25
> > > > > **Re: Re: Overfitting**
> > > > >
> > > > > Dear reviewer "PoXu" and AC,
> > > > >
> > > > > Thank you all for this thoughtful discussion!
> > > > >
> > > > > **We fully echo AC’s comments on the first point.** The notion of overfitting in supervised learning is not well defined for unsupervised setting, as there is no training or testing set. But just like the clustering or low-rank matrix completion algorithm, our model also requires a non-degenerated choice of hypothesis class to avoid trivial outcomes. For example, if we let H contain arbitrary functions, then indeed Eq. (2) can fit for any recovered qualities $y$ and achieve the minimum value 0. This is easily preventable in practice, as one should never pick $k = n$ in k-means clustering. So the real interesting question is how to pick the best H for different setups. Though this is beyond the scope of current paper, we will add a paragraph of discussion to spur follow-up studies on this particular research direction.
> > > > >
> > > > > **Now to directly answer AC’s second question,** if H is trivially chosen to be as large as the set of all functions, then our model indeed would not do any correction and thus result in even worse calibration result than Average (what is done today).  However, if we just let H be as large as all monotone functions, which is exactly our LSC-mono model, Eq. (2) will already perform quite non-trivial correction. In fact, empirically, we find the recovery performance of LSC-mono significantly better than that of Average baseline in both our synthesized dataset, where the scoring function could be linear or monotonic, and the peer grading dataset suggested by the reviewers, where the scoring function class is unknown. Finally, we remark that our framework does include the Average baseline as a special case, by picking H as identity mapping only and the distance function as L1.
> > > > >
> > > > > In summary, **the primary goal of this paper was set to propose a flexible unsupervised learning framework for the peer review calibration.** We start by assuming H is given as the set of linear functions, because (1) it is the common assumption in prior work (2) we need to properly compare LSC with various baselines with similar assumption. In future work, we wish to focus on the study of the data-dependent or the more robust choices of H.

---

### Author Response · Authors · 2021-08-10
**Global Response on the real world experimental results.**

We sincerely appreciate all the reviewers' time and their positive feedback particularly about the significance of the problem and our theoretical results. Since both the AC and Reviewer PoXu have questions about testing our algorithm on the suggested peer grading data and MTurk dataset (and  thank you for the valuable suggestions), we here respond to this particular question of real data set globally,  while respond to other questions separately to each reviewers.

We would like to start by respectfully pointing out that the suggested data sets above are not the most ideal to test our algorithms for peer review setups  due to various reasons. Specifically, for the **ICLR** or **PeerRead** dataset, the reviewers are anonymous in a way that we cannot recover the papers reviewed by the same reviewer.
We also face the exact issue mentioned by AC, that using the final decisions or average rating as ground truth since those decisions would be based on the miscalibrated scores. For the same reason, the review data in **PrefLib** pointed out by Reviewer PoXu without the ground truth rating cannot accurately evaluate the algorithms in our setting. The **MTurk** dataset suggested by the AC points out a potential way to obtain real world peer review dataset. But the currently available one by Shah et al. is difficult to utilize, because its tasks were designed to generate questions randomly for each worker, so we can hardly know which set of workers answer the same question. This prevents us from running any peer review calibration algorithm.


The peer grading dataset suggested by both AC and Reviewer PoXu is nice in the sense that we do have the grade from TA to be treated as the ground truth. However, we want to point out a fundamental difference between peer review and peer grade. That is, in peer grading, incentives and graders' strategic manipulation could play an important role and destroy the *truthfulness* of grades. In fact, much research  has been devoted to study the strategic manipulation in peer grading [1, 2], and this means an agent's scoring functions cannot be simply modeled as independent of their own grades. However, such strategic manipulation does not present in the setting of our interest, e.g., peer review and other crowdsourcing tasks (there might be some debate about peer review, but we strongly believe that most reviewers do provide objective reviews, independent of their own submissions)


Nevertheless, for the sake of our interest, we have  conducted experiments on the suggested peer grading dataset collected by the University of Hamburg to compare our algorithm with the baselines. Specifically, we now treat each homework as a "conference", the submission to each question as a "paper", each peer grade as a "review" and the averaged TA grade as the "true quality". There are six homeworks in total: each assignments involves roughly 250 submissions and 200 student reviewer; each submission have at least 6 reviews. Our algorithm does show stable performance across various parameters, especially the ranking based metrics. We list one set of results in the table below that we pick HW1 and select the top 50\% of submission for example. For more information, we attached the plots in these three links (https://ibb.co/Yy4ZMWn),  (https://ibb.co/S0mxYgj) and (https://ibb.co/fYNzmkH) that respectively measures the performance of our proposed LSC and the baselines to select the top 20\%, 50\% or 80\% of the submissions in each homework.

|       | Average | LSC (linear)     | LSC (mono) | QP | Bayesian |
| :---        |    :----:   |         :---: | :---: | :---: | :---: |
| Precision   |  0.80     |  **0.82**  |  0.78  |  0.80 |  0.78  |
| MAP    |   0.41  | 0.89   |  0.88  |  **0.90**  |  0.72  |
| NDCG    |   0.34  |  **0.85**  |  0.81  |  0.82  |   0.71  |


Despite our good performance in the above dataset, frankly speaking we still believe that our experiments on the synthetic data are still the most convincing justification of our algorithm's effectiveness.  This is probably also why all of our prior work are tested on the synthesized dataset as well. It is indeed a very intriguing open question to obtain an ideal dataset for future work of the peer review setting.  However,  this is out of the scope of the current paper.



[1] Stelmakh, Ivan, Nihar B. Shah, and Aarti Singh. "Catch Me if I Can: Detecting Strategic Behaviour in Peer Assessment."  *Proceedings of the AAAI Conference on Artificial Intelligence. Vol. 35. No. 6. 2021.*

[2] Zarkoob, Hedayat, Hu Fu, and Kevin Leyton-Brown. "Report-Sensitive Spot-checking in Peer Grading Systems." *Proceedings of the 18th International Conference on Autonomous Agents and MultiAgent Systems. 2019.*

---

### Decision · Program_Chairs · 2021-09-27

**Decision:**

Accept (Poster)

**Comment:**

This paper addresses miscalibration in peer review. Miscalibration can cause problems with the fairness of the peer review process, but so far not much has been achieved in terms of theoretical algorithms that also have practical appeal. Some previous papers study simple linear models and another study considers arbitrary or adversarial miscalibration. This paper fills in this void by considering general classes of miscalibration models. It proposes an interesting "LSC" method, provides theoretical results to back it up, and simulations to investigate it further. The rebuttal, in response to some reviewers' comments, also provides experiments on real-world data and their algorithms work well here. This is a good contribution to the field and I recommend acceptance.

Important: In their camera ready version, the authors should incorporate the various points discussed with the reviewers or with me during the review process. We hope that this review process has helped the authors improve the paper.

Summary of reviews:
- Reviewer o3ak provides a detailed review, regards the paper as novel and interesting
- Reviewer zLc7 makes some initial criticisms in their initial review. The rebuttal addresses the objective parts of these criticisms but the reviewer did not show up after the rebuttal (I had sent multiple pings).
- Reviewer PoXu provides a detailed review. They make several criticisms in their initial review, and the objective parts of these criticisms are addressed in a 3-way discussion between me, the authors, and the reviewer.
- Reviewer Emh1's review has only minor points of concern